# The Effect of Antecedents of Teachers’ Subjective Career Success

**DOI:** 10.3390/ijerph191711121

**Published:** 2022-09-05

**Authors:** Tri Muji Ingarianti, Fendy Suhariadi, Fajrianthi Fajrianthi, Ika Febrian Kristiana

**Affiliations:** 1Doctoral Program of Psychology, University of Airlangga, Surabaya 60286, Indonesia; 2Faculty of Psychology, University of Muhammadiyah Malang, Malang 65144, Indonesia; 3Faculty of Psychology, University of Diponegoro, Semarang 50275, Indonesia

**Keywords:** antecedent, career commitment, leader–member exchange, perceived organizational support, professional commitment, subjective career success, teacher

## Abstract

Career success is often seen as identical to objective matters such as high income and position. Meanwhile, one can see their success better when they build their own criteria of career success. In this regard, the present study aims to see the effect of internal (i.e., career commitment and professional commitment) and external variables (leader–member exchange and perceived organizational support) on teachers’ subjective career success. This quantitative study involved 320 teachers as participants, recruited using the accidental sampling technique. The data were collected using The Career Commitment Measure, Professional Commitment Scale, Leader–Member Exchange Multidimensionality, Survey of Perceived Organizational Support, and Subjective Career Success Inventory. The analysis result shows that career commitment, professional commitment, leader–member exchange, and perceived organizational support significantly affect teachers’ career success.

## 1. Introduction

The term career denotes a path an individual goes through related to their life-long work activities [1]. One’s career development represents a sequence of an individual’s life-long process. Teachers are strategic and significantly influential players in the field of education. According to Indonesian Law No. 14/2005 on Teachers and Lecturers, a teacher is a professional educator because any individual willing to be a teacher is required to attend formal education and have adequate skills. A teacher should possess professional, career-oriented competencies. Teachers’ professionalism plays a central and strategic role in the field of education. Teachers’ roles and responsibilities vary, ranging from educating their students to working on educational management tasks. Despite their central roles in preparing the country’s generation, rewards for their noble tasks appear to be less than they deserve. Therefore, being a teacher could be highly challenging when individuals do not have adequate self-motivation and self-esteem to achieve their career goals.

In the past few years, the construct of career success has been inseparable from the notions of objective and subjective success. While the former emphasizes objectively measured standards in evaluating one’s career success (e.g., salary, promotion, or employment status), the latter puts more emphasis on job-related and life goal achievements set based on their own criteria and evaluation [2]. In other words, individuals may have different perspectives when perceiving career success, which is usually associated with career satisfaction [3].

Teachers with subjective career success orientation tend not to see their heavy workload as burdensome hindrances [4]. Subjective career success orientation may help teachers perform their responsibility, develop competence, and achieve their career goals more optimally. Teachers with subjective career success orientation are likely to see their success as self-satisfaction (e.g., a teacher perceives success when their students gain a high score on a test; teachers are satisfied with their hard work and personal values). Teachers with professional values (e.g., idealism) may exhibit higher satisfaction and commitment than those who teach solely for economic reasons [5]. Individuals who work based on their idealism will likely have a higher intrinsic motivation to achieve career success.

Individuals may achieve their subjective career success when they understand their views and career-oriented jobs. One’s work-life balance is also reported to psychologically affect their career success [1]. Subjective career success can affect one’s self-confidence and motivation to exhibit positive experiences by developing their own resource [6]. In addition, one’s subjective career success can also be seen from job satisfaction levels. Subjective career success is associated with one’s life satisfaction [6].

Factors affecting individuals’ subjective career success vary, depending on the internal (individual) and external (organizational) contexts. Regarding the internal context, career and professional commitment affect one’s subjective career success. Career commitment denotes the extent to which an individual is motivated to work and stay in the chosen career or role [7], which comprises career planning, identity, and resilience [8]. Individuals with strong career commitments tend to be willing to finish difficult and challenging job tasks to exhibit a satisfying performance [9]. Highly committed individuals are likely to achieve their subjective career success as they do not see success based solely on external aspects.

Professional commitment serves as one of the internal factors that may lead to a teacher’s subjective career success. Professional commitment is important for developing trust and acceptance, eventually motivating individuals to perform their tasks more optimally. Professional commitment represents one’s commitment to achieving professional goals, feelings of involvement and a sense of belonging, a combination of personal and professional goals, and efforts to fulfill the job requirements [10]. Teachers’ competence and commitment relate to their attitude toward their profession, and satisfaction is one of the dimensions of subjective career success.

Regarding external aspects, leader-member exchange and perceived organizational support are known to affect one’s subjective career success. In this regard, a leader’s support is needed to ensure teachers are comfortable in the work environment. Koekemoer et al. [11] state that members tend to perceive career success when their leaders are interested in and recognize their work outcome. Employees, especially teachers, rely heavily on their leaders’ support to deliver optimal performance. Such positive support is usually termed leader–member exchange (LMX), a relationship in which leaders and teachers exhibit mutual benefits and bring advantages to the organization. Alford’s [12] study reports a positive relationship between trust-building LMX and employees’ work motivation, job satisfaction, and organizational commitment.

Perceived organizational support defines an employee’s perception of support given by their organization related to their socio-emotional well-being. Employees will likely return their organization’s support by exhibiting productivity [13]. Organizational supports can be intangible (e.g., recognition, fair treatment, work safety, autonomy) or tangible (e.g., salary, bonus, and reward). Employees believe their organization will help them face problems and forgive them when they make mistakes [14].

In this study, four hypotheses were formulated: career commitment affects subjective career success, professional commitment affects subjective career success, leader-member exchange affects subjective career success, and perceived organizational support affects subjective career success.

### 1.1. Career Commitment (CC) and Subjective Career Success (SCS)

Career commitment can be defined as employees’ engagement with their job and profession. However, employees committed to their organization are not necessarily committed to their careers. In other words, individuals with career commitment tend to focus more on their career than their organization, i.e., its working condition or coworkers. Despite high organizational job dissatisfaction, committed individuals are likely to stay with the organization to develop their careers [15]. Individuals with high career commitment tend to set high career goals and work hard to achieve them despite obstacles and drawbacks during the process [16].

Employees who are highly committed to their careers have a broader chance of career success. Subjective career success is an individual’s satisfaction with his/her career. Employees’ career success can be seen as a real achievement in one’s career and is often seen as one’s career goal. In this regard, organizations usually retain successful employees, i.e., individuals who manage to achieve their career success and exhibit performance that supports the organization [15]. Subjective career success can be seen as criteria an individual sets regarding his/her experience, measured based on one’s satisfaction with their own career.

Employees with higher career commitment tend to report more satisfying work outcomes, thus improving their SCS [17]. They are also likely to perceive positive feelings about their career progress and goal attainment. Employees with high career commitment tend to set high career goals and are assiduous in achieving success [18]. They are also well motivated to achieve their subjective career success. Employees with high career commitment often have a clear career plan and goal, making them more open to gaining knowledge that potentially leads to career success [19].

**Hypothesis** **1** **(H1).**
*Career commitment affects subjective career success.*


### 1.2. Professional Commitment (PC) and Subjective Career Success (SCS)

Professional commitment depicts an individual’s loyalty toward their career, coworkers, and professional norms they agree upon. Individuals with professional commitment often exhibit more efforts to achieve professional goals without being asked to [20]. Professional commitment is one of the important factors affecting individuals, particularly their acceptance of professional ethics and organizational goals [21]. Individuals with higher professional commitment can easily identify their professional values [22].

Affective professional commitment is a pivotal factor in an individual’s career. It is important to have a high affective professional commitment in order to maintain professional development while maintaining emotional engagement. When individuals manage to keep up with their profession, they will likely develop their professional goals and improve their performance and SCS [22,23]. Normative professional commitment is also an important factor in minimizing the adverse effect of work conflict on one of the SCS dimensions, i.e., job satisfaction [24]. Professional commitment is reported to positively affect one’s career competence [25]. Furthermore, Hasanti [26] found that career commitment significantly affects SCS. In other words, professional commitment may indirectly affect one’s SCS.

Previous works also report the effect of professional commitment on other SCS dimensions. Professional commitment is reported to improve the employees’ performance, positively influencing other coworkers and the organizations [27,28]. It potentially increases one’s positive view of their job, making them think about their family and personal life. Professional commitment can also improve one’s interest in their current job, hence improving their quality of work [29,30].

**Hypothesis** **2** **(H2).***Professional commitment affects subjective career success*.

### 1.3. Leader-Member Exchange (LMX) and Subjective Career Success (SCS)

Leader-member exchange focuses on a relationship between a leader and the member, mainly aiming at optimizing organizational success by building positive interactions between them. Sparrowe and Liden [31] state that employees with high-quality LMX can easily integrate themselves into the leaders’ personal networks. In the organizational context, teachers tend to develop an interpersonal relationship with the leader based on trust. Leaders with high LMX may provide a comfortable atmosphere in the work environment [32]. Leader-member exchange may also create a highly beneficial social network for the organization [33].

Wayne et al. [34] report that a positive relationship between leaders and their members helps teachers see their job more meaningfully and eventually positively affects their work outcome. Such a positive relationship will likely improve organizational performance (e.g., teachers exhibit higher competency, higher loyalty to the organization, and higher contribution to the organization) [35]. In this manner, teachers are likely to achieve their SCS.

A good leader–member exchange may improve employees’ perceived career success as they may enjoy formal and informal rewards from the organization. Byrne et al. [36] state that formal rewards received by teachers (i.e., training, career development, higher salary), in addition to informal rewards (i.e., positive relationships), can improve employees’ satisfaction with the leaders and their job, and eventually resulting in a higher SCS.

**Hypothesis** **3** **(H3).***Leader–member exchange affects subjective career success*.

### 1.4. Perceived Organizational Support (POS) and Subjective Career Success (SCS)

Perceived organizational support is known to relate to SCS. The concept of POS refers to employees’ perception of the organization’s support and direction for finishing tasks effectively or coping with stressful conditions and perceptions that their organizations listen to them and care about their emotional well-being [37]. Employees with high POS are likely to achieve career success. Perceived organizational support is helpful for individuals in achieving their career success [38] because it promotes various factors believed to improve employees’ SCS, such as job achievement, satisfaction, and commitment.

Every individual possesses a career goal and works hard to achieve it; yet, as social creatures, individuals need support from their surroundings. This support can come from coworkers, supervisors, or the organization itself in the work environment. Supervisor support, e.g., appreciation and career growth guidance, may positively affect individuals’ careers. In addition to supervisor support, organizational rewards and work conditions also play important roles in determining employees’ satisfaction with their work outcomes. Employees’ career satisfaction is related to broader contexts. In the performance satisfaction–effort loop, a causal relationship stemming from a good performance could bring economic, social, and psychological rewards for employees. In this regard, the organization’s fair reward system will likely result in employee career satisfaction.

Employees who perceive support from their organizations tend to improve their performance and loyalty toward the organization. Employees’ perceived organization support may encourage them to return what they have received from the organization. In this manner, organizations will likely benefit from higher employees’ work commitment, effective performance, and lower turnover, while employees perceive higher emotional support, affiliation, self-esteem, and agreement, thus creating a positive work environment and improving their subjective career success [39]. As asserted by Rhoades and Eisenberger [37], employees perceiving support from their organization tend to see meaningful values at work. Therefore, perceived organizational support is viewed as a predictor of employees’ SCS. All hypotheses are illustrated in Figure 1.

**Hypothesis** **4** **(H4).**
*Perceived organizational support affects subjective career success.*


## 2. Research Methodology

### 2.1. Research Design

The present study applied a quantitative method—a method used to study a certain population or sample in order to test hypotheses [40]. For the purpose of this study, a correlational design was applied with simple and multiple linear regression tests to see the effect of independent variables on the dependent variables. This study examined five variables. The X variables consisted of career commitment, professional commitment, leader-member exchange, and perceived organizational support. Meanwhile, variable Y in this study was the subjective career success.

### 2.2. Participants

A random purposive sampling technique was employed due to the large number of population members. Individuals deemed to fit the population characteristics and willing to participate in this study were recruited as participants, as evidenced by signing the informed consent form. The number of samples was determined a priori by considering: statistical power (>0.80), 5% error degree, number of predictors (*n* = 4), expected effect size (0.30), and type of multiple regression random model test with the direction of the one-tailed hypothesis. Using the G-Power program, the recommended minimum sample size was determined to be 36 participants.

The study participants were 320 teachers recruited using a random purposive sampling technique. They were Indonesian teachers with at least five years of working experience in public and private schools of various educational levels and held bachelor’s or associate’s degrees following Law No. 14 of 2005 on Teachers and Lecturers. They also held a decree of appointment issued by their institutions.

### 2.3. Research Variables and Instruments

Career commitment was measured using Carson and Bedeian’s [8] Commitment Career Measure (CCM), which has been adapted by Ingarianti et al. [41]. One of the items read, “The field I work in is an important part of my identity.” Meanwhile, professional commitment was measured using the Professional Commitment Scale developed by Meyer et al. [42] and adapted by Smith and Hall [43] (e.g., “It would be costly to change my profession now”).

The leader–member exchange was measured using LMX-MDM developed by Liden and Maslyn [33]; one of the items read, “I work for my leader beyond my job description”. Employees’ perceived organizational support was measured using the Survey of POS (SPOS) developed by Eisenberger et al. [44] and revised by Eisenberger et al. [45]. One of the items reads, “My organization takes my goal and values into consideration”. The subjective career success was measured using the Subjective Career Success Inventory (SCSI) developed by Shockley et al. [46] and adapted by Ingarianti et al. [47], containing 24 items with 3 items measuring each dimension. One of the items reads, “I am sure my job brings changes to the organization”. This study has obtained permission from the ethics commission or institutional review board (IRB), number 2396-KPEK.

### 2.4. Procedure and Data Analysis

This study consisted of three main stages: The first was preparation, in which we deepened our understanding of the issue and theoretically examined dependent and independent variables (SCS and career commitment, respectively). After that, we prepared the measure for each variable through international journal articles and studies on the adapted measures. The research instrument was adapted into Indonesian to suit the population/background of the participants. The scale was distributed to 320 respondents between November and December 2021. The collected data were analyzed using simple and multiple linear regression tests with SPSS 26. Lastly, the conclusion was drawn based on the result of the study.

## 3. Results

This study involved 320 respondents who suited the determining criteria. The following Table 1 displays the participants’ demographic backgrounds.

As shown in the table, most participants were female teachers (66.9%). Most participants were known to live in Java (87.5%) and hold bachelor’s degrees (92.2%). Most participants were civil state apparatuses (47.9%), subject teachers (69.7%), and worked in senior high school (37.2%). Most participants have worked for more than 15 years (51.9%) and held a teacher certificate (68.1%), with a monthly income of less than IDR 5 million (77.8%).

As shown in Table 2, all dimensions in this study exhibited a significant result (*p* < 0.05).

Table 3 displays a significant analysis result (*p* < 0.05). Taking a closer look at each dimension, affective and normative professional commitments were found to affect quality work.

The multiple linear regression test results shown in the Table 4 above demonstrate a significant relationship between LMX and SCS (*p* < 0.05). In other words, LMX affects each SCS dimension. Furthermore, the affective dimension exhibits a significant role in all SCS dimensions, and contribution plays an important role in recognition. No significant result was found in other dimensions, as the regression coefficient was higher than 0.05.

The simple linear regression test result displayed in Table 5 shows that POS plays a significant role in eight SCS dimensions (*p* < 0.05). Employees’ perceived organizational support was found to affect teachers’ meaningful work and influence.

## 4. Discussion

### 4.1. Career Commitment and Subjective Career Success

The discussion is presented in eight sections based on the effect of career commitment on SCS dimensions. First, career identity was found to affect recognition. In this regard, teachers’ career identity increases their sense of attachment to their profession. Individuals committed to their careers are likely to exhibit more engagement with their organization, job, and career to keep growing despite obstacles they face, making them be recognized for their job [48]. Second, career identity was also found to significantly affect the quality of work. In this regard, teachers’ career identity makes individuals perceive higher quality performance. This is in line with Herachwati and Rachma [49], who state that individuals with higher career commitment tend to make more effort to achieve their career target and satisfaction.

Third, career commitment was found to significantly affect the dimension of meaningful work. In other words, individuals with higher commitment can be recognized more easily when involved in their job. Career commitment is also found to significantly affect SCS in Mahendra’s [50] study. Fourth, career identity and resilience were found to significantly affect the dimension of influence. In this regard, individuals perceiving an influence are likely to see themselves making an important contribution to the organization and their social environment. In other words, individuals with higher career commitment may perceive success when they can bring impact to others or their organization.

Fifth, career commitment was found to significantly affect authenticity. Individuals with higher career commitment tend to be motivated by their career hope and goal. In other words, career commitment plays a pivotal role in predicting one’s career success. Sixth, career identity was found to affect personal life. Individuals can have a career with positive impacts on others. Thus, individuals with career identities may see themselves as useful individuals.

Seventh, career commitment was also found to affect growth and development, meaning that individuals with higher career commitment are capable of developing their careers by looking at and extending their career-related knowledge. A previous study conducted by Srikanth and Israel [51] report the effect of career commitment on employees’ career success. Their study showed that individuals with higher career commitment would likely work harder to achieve their career goals. Eighth, career identity and career planning were found to significantly affect the dimension of satisfaction. In other words, individuals with proper career identity and planning are likely to have more positive feelings and satisfaction toward their careers. Individuals with higher career commitment tend to exhibit more positive feelings related to their career achievement and career success compared to those with lower career commitment [52].

### 4.2. Professional Commitment and Subjective Career Success (SCS)

As shown in Table 3, teachers with higher affective and normative professional commitments tend to report a higher quality of work and lower intention to quit. This finding supports Zhao et al. [53], who found that individuals with higher professional commitment exhibit higher quality of work. Affective and normative professional commitments were also found to affect the dimension of meaningful work. Perceived meaningful work is a psychological condition that directs toward a positive work outcome. In this context, teachers will likely focus more on obtaining emotional meaningfulness and engagement with their profession or organizations. A good relationship between employees and their leaders can stimulate autonomy in performing the given task, thus creating meaningful work and substantially improving teachers’ professional commitment. In the same vein, Khan and Nemati [54] view that affective and normative professional commitments possess a stronger relationship with meaningful work when compared to continuance professional commitment.

Continuance and normative professional commitments were found to affect the dimension of influence. In this regard, teachers may perceive higher satisfaction and perceive subjective success by helping their coworkers achieve their goals. Teachers’ participation in various professional skills plays pivotal roles in their profession. Teachers with higher professional commitment tend to make a positive contribution to other individuals and their profession. The professional commitment represents one’s willingness to contribute and devote themselves to their profession [55]. That is, teachers are considered to achieve subjective career success when they can positively contribute to other individuals and professions.

Responsible, professionally committed teachers will likely build a career path that suits their personal needs. This is supported by the multiple linear regression test result, where affective and normative professional commitments were found to affect teachers’ authenticity. Teachers who are committed to their profession may obtain support from others and establish their own career path, which can eventually affect their subjective career success and exhibit positive professional values and behaviors. Teachers with low attachment to others are likely to exhibit more consistent behavior related to their professional commitment and belief, thus establishing career authenticity. This finding supports Cho and Huang [56], who found that individuals with high affective professional commitment tend to exhibit higher career authenticity.

Affective and continuance professional commitments were also found to affect one’s career growth and development. Teachers who are willing to achieve their professional goals tend to be more aware of their career development by learning new skills and knowledge. On the contrary, teachers who perceive a high cost of leaving their current profession tend to be unaware of their career growth. The professional commitment may improve teachers’ skills because when they are involved and satisfied with their work, they are likely to exhibit their best to develop their profession and career. In this regard, they will be motivated to perform the given task optimally as they perceive that their personal value equals their professional ones. They are likely to be proud when they can finish their professional responsibility. This supports Chow’s [57] study, which reports that employees are required to improve their professionalism through continuance professional commitment.

### 4.3. Leader–Member Exchange (LMX) and Subjective Career Success

As displayed in Table 4, teachers with a higher LMX are likely to receive support and appreciation or recognition from their leaders. Leaders’ recognition can make teachers feel appreciated and perceive a higher career satisfaction. With high career satisfaction, teachers will likely evaluate their careers positively and achieve their subjective career success. Leaders’ support can also help teachers determine their career goals and eventually perceive career satisfaction. This is in line with Breevaart et al. [58], who found that LMX is affected by leaders’ support, where leaders may positively affect the teachers’ perception of their career satisfaction and evaluation.

A significant result was noticed in the relationship between contribution and recognition dimensions (*p* < 0.05). This result indicates that when teachers give an optimal contribution to their job, they will likely receive recognition from their leaders, which allows them to subjectively evaluate their careers [59].

A significant relationship was also noticed between the affective dimension and all SCS dimensions (*p* < 0.05). This result shows that when a leader builds a positive relationship with teachers, they may have a good leader–member relationship. Such a positive relationship can make teachers feel recognized or appreciated. Positive relationships with leaders can help teachers perceive their subjective career success [60].

### 4.4. Perceived Organizational Support (POS) and Subjective Career Success (SCS)

Table 5 shows that organizational support may create a mutually beneficial relationship between teachers and the organization. In this regard, teachers will likely exhibit their best performance to deliver a quality work outcome. Individuals perceiving a high POS tend to contribute more to the organization [61].

The influence dimension exhibited higher scores than other dimensions, as teachers perceived sincere support from their organization. Organizational support was also found to affect the recognition dimension. In other words, teachers perceived the organization’s fair treatment and equal opportunities to grow. The organization’s fair reward system can improve employees’ performance and self-esteem [62], as teachers find their contributions are appreciated and recognized by the organization and coworkers.

Further, POS was also found to affect teachers’ career satisfaction. Organizational support (e.g., appreciating the contribution, listening to teachers’ problems, and appreciating the achievement) may result in teachers’ satisfaction with their careers. Organizational support can affect their career satisfaction, substantially influencing their career perception [63]. This is in line with Puspitasari and Ratnaningsih [64], who state that individuals with positive emotions in their workplaces are likely to positively influence their non-work environment. This occurs because teachers see themselves as capable of managing their time and responsibility at work, which eventually affects their non-work life [64]. Personal life exhibited the lowest score as it is affected by non-work factors. However, POS was found to affect teachers’ personal life.

### 4.5. Practical Implication

Teachers are expected to have and develop their subjective career success to be more emotionally attached to their career and more dedicated to improving their subjective career success. Because subjective career success is related to past work experience and orientation towards career progress, the practical implication that can be suggested is that teachers can find and reinforce positive experiences (emotional, cognitive, and behavioral) in carrying out their work. Positive job-related experiences will likely strengthen their subjective career success. Teacher capacity development programs are also necessary to help teachers exhibit beyond-expectation performance.

Future studies are expected to garner the data using a qualitative approach through interviews to obtain a more comprehensive picture of the phenomenon. Research with a mixed-method design will be able to enrich the data obtained and better describe teachers’ SCS and its underlying factors. It is also necessary to conduct a similar study on other professions, considering that subjective career success may change in every career stage. This study can be used as a reference in the field of psychology related to subjective career success.

### 4.6. Limitation and Novelty

Despite the present study’s novelty, some limitations need to be discussed. This study was only conducted in East Java and Kalimantan, not all Indonesian regions. However, as existing studies involve mostly blue-collar workers, studies on antecedents of subjective career success among teachers could be seen as a novelty this study proposes. This study also provides a per-dimensional analysis, describing the effect of antecedent variables on teachers’ subjective career success. Furthermore, subjective career success is dynamic and ever-changing in each career stage, limiting the conclusion drawn from this study.

## 5. Conclusions

This study concludes that antecedent variables significantly affect teachers’ subjective career success. The internal variables, i.e., career commitment and professional commitment, are found to significantly affect teachers’ subjective career success. Meanwhile, the external variables, i.e., leader–member exchange and perceived organizational support, significantly affect teachers’ subjective career success. In other words, the proposed hypotheses in this study are accepted. From the two groups of antecedent variables for teachers’ career success, internal antecedents show a higher correlation effect than external antecedents. These findings indicate that career success in teachers is personal and has implications for the importance of paying attention to the psychological condition of teachers so that they can achieve a successful career. However, longitudinal research needs to be done to prove that internal antecedents have a big role in teachers’ career success.

## Figures and Tables

**Figure 1 ijerph-19-11121-f001:**
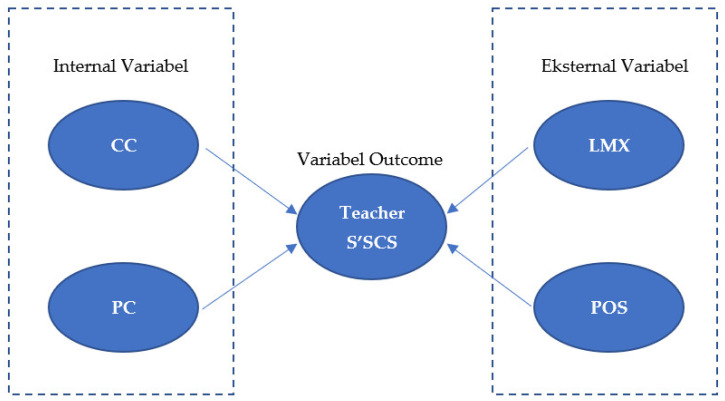
Description: career commitment (CC); professional commitment (PC); leader–member exchange (LMX); perceived organizational support (POS).

**Table 1 ijerph-19-11121-t001:** Participants’ demographics.

Category	Frequency	Percentage (%)
Gender		
Male	106	33.1%
Female	214	66.9%
Domicile		
Java	280	87.5%
Non-Java	40	12.5%
Educational Background		
Associate Degree/4-Year Diploma	6	1.9%
Bachelor Degree	295	92.2%
Master Degree	19	5.9%
Employment Status		
Civil State Apparatus Teacher	150	46.9%
Organizational Permanent Teacher	89	27.8%
Contract Teacher	81	25.3%
Work Unit		
Kindergarten	31	9.7%
Elementary School	62	19.4%
Junior High School	108	33.8%
Senior/Vocational High School	119	37.2%
Length of Service		
Less than 10 years	100	31.2%
10–15 years	73	22.8%
More than 15 years	147	45.9%
Educator Certificate		
Have	218	68.1%
Do not have	102	31.9%
Income		
Less than IDR 5 million	249	77.8%
More than IDR 5 million	71	22.2%

**Table 2 ijerph-19-11121-t002:** Multiple linear regression test of career commitment and subjective career success.

Variable Relationship	R *Square*	ANOVA Sig.	*Coefficient*	
			Β	Sig.
CI → R	0.217	0.000	0.339	0.000
CP → R	0.217	0.000	0.033	0.498
CR → R	0.217	0.000	−0.040	0.321
CI → QW	0.105	0.000	0.297	0.000
CP → QW	0.105	0.000	0.099	0.091
CR → QW	0.105	0.000	−0.092	0.109
CI → MW	0.142	0.000	0.329	0.000
CP → MW	0.142	0.000	0.145	0.012
CR → MW	0.142	0.000	−0.120	0.033
CI → I	0.91	0.000	0.290	0.000
CP → I	0.91	0.000	0.090	0.126
CR → I	0.91	0.000	−0.139	0.016
CI → A	0.164	0.000	0.300	0.000
CP → A	0.164	0.000	0.230	0.000
CR → A	0.164	0.000	−0.108	0.053
CI → PL	0.61	0.000	0.243	0.000
CP → PL	0.61	0.000	0.039	0.519
CR → PL	0.61	0.000	−0.071	0.229
CI → GD	0.175	0.000	0.307	0.000
CP → GD	0.175	0.000	0.245	0.000
CR → GD	0.175	0.000	−0.151	0.006
CI → S	0.167	0.000	0.306	0.000
CP → S	0.167	0.000	0.229	0.000
CR → S	0.167	0.000	−0.104	0.061

Description: CI: career identity, CP: career planning, CR: career resilience, R: recognition, QW: quality work, MW: meaningful work, I: influence, A: authenticity, PL: personal life, GD: growth and development, S: satisfaction.

**Table 3 ijerph-19-11121-t003:** Multiple linear regression test of professional commitment and subjective career success.

Variable Relationship	R *Square*	ANOVA Sig.	*Coefficient*	
			Β	Sig.
APC → R	0.118	0.000	0.112	0.089
CPC → R	0.118	0.000	0.086	0.138
NPC → R	0.118	0.000	0.223	0.002
APC → QW	0.124	0.000	0.179	0.007
CPC → QW	0.124	0.000	0.088	0.129
NPC → QW	0.124	0.000	0.174	0.014
APC → MW	0.165	0.000	0.213	0.001
CPC → MW	0.165	0.000	0.035	0.532
NPC → MW	0.165	0.000	0.228	0.001
APC → I	0.100	0.000	0.078	0.241
CPC → I	0.100	0.000	0.134	0.022
NPC → I	0.100	0.000	0.187	0.009
APC → A	0.209	0.000	0.298	0.000
CPC → A	0.209	0.000	0.074	0.179
NPC → A	0.209	0.000	0.181	0.007
APC → PL	0.063	0.000	0.110	0.106
CPC → PL	0.063	0.000	0.023	0.698
NPC → PL	0.063	0.000	0.160	0.029
APC → GD	0.159	0.000	0.308	0.000
CPC → GD	0.159	0.000	0.122	0.031
NPC → GD	0.159	0.000	0.070	0.310
APC → S	0.118	0.000	0.264	0.000
CPC → S	0.118	0.000	0.077	0.181
NPC → S	0.118	0.000	0.079	0.268

Description: APC = affective professional commitment; CPC = continuance professional commitment; NPC = normative professional commitment; R = recognition; QW = quality work; MW = meaningful work; I = influence; A = authenticity; PL = personal life; GD = growth and development; S = satisfaction.

**Table 4 ijerph-19-11121-t004:** Multiple linear regression test of leader–member exchange and subjective career success.

Variable Relationship	R *Square*	ANOVA Sig.	*Coefficient*	
			Β	Sig.
CON → R	0.125	<0.001	0.296	<0.001
LOY → R	0.125	<0.001	0.035	0.565
AFF → R	0.125	<0.001	0.171	0.004
PC → R	0.125	<0.001	−0.092	0.227
CON → QW	0.142	<0.001	0.117	0.119
LOY → QW	0.142	<0.001	0.089	0.146
AFF → QW	0.142	<0.001	0.270	<0.001
PC → QW	0.142	<0.001	8.520	0.991
CON → MW	0.095	<0.001	0.099	0.198
LOY → MW	0.095	<0.001	−0.062	0.323
AFF → MW	0.095	<0.001	0.247	<0.001
PC → MW	0.095	<0.001	0.048	0.533
CON → I	0.154	<0.001	0.139	0.061
LOY → I	0.154	<0.001	0.156	0.011
AFF → I	0.154	<0.001	0.282	<0.001
PC → I	0.154	<0.001	−0.093	0.211
CON → A	0.089	<0.001	0.048	0.530
LOY → A	0.089	<0.001	−0.098	0.120
AFF → A	0.089	<0.001	0.178	0.004
PC → A	0.089	<0.001	0.171	0.028
CON → PL	0.050	0.003	0.057	0.468
LOY → PL	0.050	0.003	0.055	0.395
AFF → PL	0.050	0.003	0.146	0.019
PC → PL	0.050	0.003	0.30	0.701
CON → GD	0.105	<0.001	0.062	0.421
LOY → GD	0.105	<0.001	0.008	0.904
AFF → GD	0.105	<0.001	0.266	<0.001
PC → GD	0.105	<0.001	0.044	0.568
CON → S	0.094	<0.001	0.105	0.171
LOY → S	0.094	<0.001	0.015	0.816
AFF → S	0.094	<0.001	0.214	<0.001
PC → S	0.094	<0.001	0.044	0.571

Description: dependent variables recognition, quality work, meaningful work, influence, authenticity, personal life, growth and development, satisfaction.

**Table 5 ijerph-19-11121-t005:** Multiple linear regression test of perceived organizational support and subjective career success.

Variable Relationship	R *Square*	ANOVA Sig.	*Coefficient*	
			Β	Sig.
POS → R	0.214	0.000	0.463	0.000
POS → QW	0.181	0.000	0.426	0.000
POS → MW	0.132	0.000	0.364	0.000
POS → I	0.224	0.000	0.473	0.000
POS → A	0.114	0.000	0.337	0.000
POS → PL	0.093	0.000	0.305	0.000
POS → GD	0.130	0.000	0.361	0.000
POS → S	0.125	0.000	0.353	0.000

Description: K: fairness, SS: supervisor support, ORJC: organizational reward and job conditions, R: recognition, QW: quality work, MW: meaningful work, I, influence, A: authenticity, PL: personal life, GD: growth and development, S: satisfaction, POS: perceived organizational support.

## Data Availability

The datasets generated for this study are available upon request from the corresponding author.

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
