# Peer review of "The Effect of Antecedents of Teachers’ Subjective Career Success"

_ijerph, 2022, doi:10.3390/ijerph191711121_

Round 1

Reviewer 1 Report

I am happy to review this manuscript. Authors have done good work, but there are still changes are required, that should be addressed in the revised manuscript. 

1. The introduction section should be revised, I suggest authors to add some previous research findings and explain the contribution of your study in introduction section. The novelty of the article is missing. I want to see the contribution.

2. There is no conceptual model figure in the manuscript. I suggest authors to add conceptual model figure, so I can easily understand the relationships. 

3. I want to see more data in the methodology section, this is not enough, What is the criteria for selection of population and sample. What is the sampling technique ?

4. What is the language process used for gathering a data?

5. There is no information regarding the scale rating.

6. Does authors have applied common method bias test on data?

7. The practical implication section should be revised and add more explanation. 

Good Luck

Author Response

Dear reviewers 1,

Thank you for providing excellent input for my article. Some of the things I changed are as follows:

1. Contributions have been added
2. I need extra time to make the model figure, so may I ask for some time to finish it.
3. This study uses a purposive sampling technique, the explanation is in the article.
4. data collection using Indonesian language
5. I will add
6. Explanation of implication has been added.

Thank you

Best regards,

Tri Muji Ingarianti

Reviewer 2 Report

Although the paper is interesting I believe it needs to be revised and restructured for several reasons:

1. TITLE. The word ‘antecedents’ in the title is not self-explanatory. This is not my article but I would honestly suggest using some of the key concepts such as the following: ‘’Variables affecting Teachers’ Subjective Career Success’, ‘The effect of internal and external variables on Teachers’ Subjective Career Success’, or even the more specific ‘The effects of Commitment, Leader-member Exchange and Organization Support on Teachers’ Subjective Career Success’.  Any of those alternative titles would be more indicative of the type of research.

2. KEYWORDS. As with the title, the authors should include key concepts related with the research: teachers, subjective career success, commitment, leader-member exchange, organization support. The title and the keywords are essential and should summarize the research content as they did in the abstract.

After reading twice the manuscript I believe the content could be better organized for clarity reasons:

3. INTRODUCTION. I suggest reordering paragraphs second (lines 34-43) and third (lines 44-50), just change the order. The reason is that the third paragraph somehow defines the meaning of ‘teachers’ objective and subjective success’, which should definitely be included before explaining the teachers’ perceptions.

4. INTRODUCTION. This section is too long and does not clearly state the research need and hypotheses. I recommend identifying and clearly stating the research need/objectives before in line 90, right before subsection 1.1. In fact, the following subsections 1.1, 1.2, etc are the research hypotheses but they are not clearly enunciated until the end which can be rather confusing for the readers.

5. HYPOTHESES or OBJECTIVES. The paragraph starting in line 92 is where the authors define and explain the different variables they are going to examine. Therefore, this could be reorganized as an independent section entitled ‘Objectives’ or ‘Hypotheses’ to make them clear and separate them from the introduction.  Additionally, the titles of these subsections should be carefully revised as they refer to the dependent and independent variables. For example, 1.1 (line 91 and ss.) ‘Career commitment and Subjective Career Success’, add here the abbreviated form SCS instead of doing so in the following subsection 1.2 (line 115). So, it should read as ‘Career commitment (CC) and Subjective Career Success (SCS)’ and be consistent with all the following sections throughout the paper. In the later analysis and tables, these abbreviated forms representing the variables are frequently used, so the reader might appreciate more clarity in this section. The second stands now as ‘1.2. Professional Commitment and Subjective Career Success (SCS)’ (line 115) and this could be renamed as ‘Professional Commitment (PC) and Subjective Career Success (SCS)’.  The third one is ‘1.3. Leader-Member Exchange (LMX) and Subjective Career Success’, so it could be better rephrased as ‘1.3. Leader-Member Exchange (LMX) and Subjective Career Success (SCS)’, including also the abbreviated form. Clarity and coherence are essential for the quality of the paper.

6. HYPOTHESES. As previously mentioned, each subsection included in the introduction ends with an independent sentence which is actually a research hypothesis. I would recommend moving all those hypotheses together and put them at the end of the introduction. See point 4 above. As it stands now, the titles of those subsections and the hypotheses are basically the same, so why not use the hypotheses as the titles and avoid repeating them? This can be quite confusing for the readers. The authors should clarify their objectives before and do not wait until lines 195-195 and lines 203-205 (see next comment)

7. RESEARCH DESIGN. In this section the authors state ‘a correlational design was applied with simple and multiple linear regression tests to see the effect of independent variables on the dependent variable’. Which one/s are the independent and the dependent variable/s? They should identify and make them clear before.  In RESEARCH VARIABLES AND INSTRUMENTS, the authors finally explain that ‘This study examined five variables, the variables X consists of career commitment, professional commitment, leader-member exchange, and perceived organizational support. Meanwhile, variable Y in this study was the subjective career success’ (203-204). I recommend moving this sentence to the ‘research design’ and keeping this subsection just for ‘research INSTRUMENTS’.

8. RESEARCH INSTRUMENTS (lines 202-221). This subsection should be better explained. The author/s mention the validated scales they used in the research but the readers may not be familiar with all those scales, therefore an Annex with the scales or a short description about the number of items, dimensions, etc might be very helpful. The author/s just provide one item for each scale but this does not suffice.

9. PROCEDURE AND DATA ANALYSIS. This section does not contain any ‘data analysis’, therefore it might be better simplified as ‘Research procedure’. The author/s mention three stages but these remain unclear. They should explain the meaning of ‘we prepared the measure for each variable through international journal articles and studies on the adapted measures’ (225-226). Which journals? They should also revise the content and rephrase some statements to avoid contradictions. For example, ‘we deepen our understandings of the issue and theoretically examines the dependent and independent variables (SCS and career commitment, respectively).’ (lines 223-224) So, there was one dependent and one independent variable? They stated before that there was one dependent variable (SCS) and FIVE independent variables (commitment, organization support, etc). They should definitely clarify all these issues, be consistent with the abbreviated forms and enunciation of hypotheses.

10. RESULTS. As the context (teachers in Indonesia) was not explained in the paper the international reader may not understand part of the content, for example Table 1 including descriptives in Educational Background ‘D4’, ‘S1’, ‘S2’, What do they stand for? Or in ‘Employment status’ the references ‘ASN’ and ‘organizational permanent’, then ‘teacher’ and ‘contract teacher’, what do they mean? What are the differences? Provide some context.

11. TABLES. For clarity reasons, I would suggest renaming the tables according to the hypotheses, not the type of analysis. For example, table 2 reads as ‘Table 2. Linear Regression Test Result’ (line 239), but about what? It later becomes obvious these results are for ‘Career Commitment’, after reading table 3 as ‘Table 3. Multiple Linear Regression Test of Professional Commitment and SCS. The authors should be consistent and coherent throughout the paper for clarity reasons, check back the use of abbreviated forms in reference to the variables as explained before. Besides, there is no need to include ‘significant’ and ‘not significant’ in each table (last column) if the p value (significance) is provided (‘p < .05).’ Experienced readers will be able to interpret the p value. The last column is not necessary.

12. DISCUSSION.  The author/s state ‘As shown in table 2, regarding per-dimensional analysis, we divide the discussion into eight sections’ (line 269). Which sections/dimensions? Names? Read before comment number 8 about instruments. There is no annex with the scales (number of items, names of sections/dimensions, etc)

13. PRACTICAL IMPLICATIONS. (line 391 and ss.) This subsection actually deals with FURTHER RESEARCH more than implications.

14. LIMITATIONS AND NOVELTY (line 398). As previously explained, the ‘novelty’ should be stated before.

15. CONCLUSIONS. This section is quite weak as it is based on generalizations

16. Wording. The author/s should revise the paper, some statements can be confusing, for example ‘Third, career commitment is found to significantly affect meaningful work,’ (line 279) and three lines below ‘Commitment career is also found to significantly affect SCS’ (line 281). I guess they both refer to Career Commitment or CC.

The paper requires English edition.

Author Response

Dear reviewers 2,

Thank you for providing excellent input for my article. Some of the things I changed are as follows:

1. I tried to summarize my title so that it still uses antecedents
2. keywords have been fixed
3. We have tried to improve the introduction and hypotheses
4. We have corrected the results, discussions and conclusions that have not been optimal.

Thank you

Best regards,

Tri Muji Ingarianti

Reviewer 3 Report

The topic of research is actual up today. Teachers’ professionalism plays a central and strategic role in the field of education. In the last recent years, the construct of career success is inseparable from the notions of objective and subjective success. Professional commitment serves as one of the internal factors that may lead to a teacher’s subjective career success. 

Proposals for improvement

1. The introduction is combined with a review of the literature, which does not allow the authors to formulate the main concept of the research.

2. In the introduction, 4 research hypotheses are identified. It would be advisable to single out the hypotheses model in a separate section and build a graphical model of their study and interaction.

3. The Research Design section needs to be expanded to understand research algorithms.

4. The conclusion does not reflect the results of the study at a conceptual level. The conclusion needs to be rewritten.

5. Authors apply career-oriented competencies. But the list of competencies for teachers' career development does not present.

6. It is necessary to edit the text and format of the article carefully.

Author Response

Dear reviewers 3,

Thank you for providing excellent input for my article. Some of the things I changed are as follows:

1. We have revised the introduction
2. The research design has also been
3. The conclusion is still in the process of being improved.

Thank you

Best regards

Round 2

Reviewer 1 Report

Dear Authors, I feel happy to review the revised version of this manuscript. Authors have carefully addressed all the suggestions. Therefore, I accept this manuscript for publication.  

Author Response

Dear Reviewer 1, 

Thank you for the great feedback and acceptance of this article.

Kind regards,

Tri Muji Ingarianti

Reviewer 2 Report

Dear author,

thanks for your revised version and taking into consideration my comments.

I still have some problems with the title. I understand you want to keep the word 'antecedents'. However, I believe that the title as it stands now is ambiguous as it seems to be a descriptive work. I recommend adding some other terms to your title for clarity reasons such as 'The effect of antecedents on Teachers' Subjective Career Success'   rather than 'Antecedents of Teachers’ Subjective Career Success'. This would probably clarify the research objectives and content to your potential reader. But this is your title so you can give another thought to it.  

Author Response

Dear Reviewer 2, 

I accept feedback and make changes to the title of this article. Thank you for the great feedback and acceptance of this article.

Kind regards,

Tri Muji Ingarianti

Reviewer 3 Report

The topic of research is actual. Authors use modern models for simulating teachers' career success.

Proposals

1. It is advisable to present the proposed hypothesis in the form of a diagram according to its impact on career success.

2. The second section should be called "Research Methodology"

3. The conclusions need to be expanded on the hypotheses and their adequacy. At the same time, it is expedient to show the development of the proposed models and methods.

4. In the Abstract need to be removed all abbreviations.

5. Keywords need to be extended

Author Response

Dear Reviewer 3,

I accept input and make changes to this article. Some of the things I changed:

1. Presenting hypotheses in diagram form according to their impact on career success.
2. Changed writing in the second part (Research Methodology)
3. Extending the conclusions and showing the development of the proposed model and method.
4. Eliminate all abbreviations in the Abstract.

Thank you for the great input and acceptance of this article.

Kind regards,

Tri Muji Ingarianti
